# On the Impacts of the Global Sea Level Dynamics

**Costas Varotsos** [1,2,*] , **Yuri Mazei** [1,3,4], **Nicholas V. Sarlis** [5] , **Damir Saldaev** [1,3] **and Maria Efstathiou** [1]

1 Faculty of Biology, Shenzhen MSU-BIT University, Shenzhen 518172, China; mefstathiou2014@gmail.com
2 Climate Research Group, Division of Environmental Physics and Meteorology, Faculty of Physics, National and Kapodistrian University of Athens, 15784 Athens, Greece
3 Department of General Ecology and Hydrobiology, Lomonosov Moscow State University, Leninskiye Gory, 1, Moscow 119991, Russia; yurimazei@mail.ru (Y.M.); k-brom@yandex.ru (D.S.)
4 Severtsov Institute of Ecology and Evolution, Russian Academy of Sciences, Leninskiy Ave. 33, Moscow 117071, Russia
5 Section of Condensed Matter Physics, Department of Physics, National and Kapodistrian University of Athens, 15784 Zografos, Greece; nsarlis@phys.uoa.gr
* Correspondence: covar@phys.uoa.gr

**Abstract:** The temporal evolution of the global mean sea level (GMSL) is investigated in the present analysis using the monthly mean values obtained from two sources: a reconstructed dataset and a satellite altimeter dataset. To this end, we use two well-known techniques, detrended fluctuation analysis (DFA) and multifractal DFA (MF-DFA), to study the scaling properties of the time series considered. The main result is that power-law long-range correlations and multifractality apply to both data sets of the global mean sea level. In addition, the analysis revealed nearly identical scaling features for both the 134-year and the last 28-year GMSL-time series, possibly suggesting that the long-range correlations stem more from natural causes. This demonstrates that the relationship between climate change and sea-level anomalies needs more extensive research in the future due to the importance of their indirect processes for ecology and conservation.

**Keywords:** sea level; detrended fluctuation analysis; scaling dynamics; fractal; self-similarity; long-range correlations





## 1. Introduction

Many studies have argued that observational data show that the global climate is warming up and that this phenomenon is accompanied by rising sea levels, which obviously has a significant impact on island and coastal communities. It is believed that the cause of sea level rise is mainly thermal expansion, to which the loss of both glaciers and ice sheets in Greenland and Antarctica contributes significantly. In this regard, Thomas et al. [1] suggested that the combination of ocean thermal expansion and increased loss of ice mass from Greenland is causing the acceleration of sea level rise since the 1970s. To date, there is no model or series of models that perform robust simulations in all modes of climate variability and its impacts (e.g., [2–7]).

Although a consensus has been achieved between several studies on the likely increase in GMSL at an accelerated rate, the magnitude of this acceleration needs further investigation because it is strongly dependent on the future of the two large ice sheets. In particular, one of the current "hot topics" is the projections of the GMSL rise by the year 2100. Hu and Bates [8], studying the global average and the regional sea level, found a statistically significant reduction in sea level rise between 2061 and 2080.

They have also found that there are areas where the reduction is insignificant (such as the Philippines and west of Australia) owing to ocean dynamics and the intensification of internal variability due to external forcings.

In connection with this, Vousdoukas et al. [9] predicted a likely increase of the global average extreme sea level in the period 2000–2100 by 34–76 cm. Nerem et al. [10] estimated

that the GMSL could rise $65 \pm 12$ cm by 2100 compared to 2005, which is roughly in line with the IPCC 5th Assessment Report (AR5) model projections.

It is noted that earlier, Nerem et al. [11] discussed the correlation between the ENSO (El Niño Southern Oscillation) event in the period 1997–1998 and the extreme fluctuation of GMSL. They also observed similar sudden variations for each major ENSO event since 1981, indicating ENSO's contribution to the GMSL variability.

Church and White [12] studied the linear trends of the GMSL using satellite altimeter data (from 1993–2009) and coastal and island sea-level measurements (from 1880–2009). Upward trends were detected in the first and second datasets, with sizes of $3.2 \pm 0.4$ mm per year and $2.8 \pm 0.8$ mm per year, respectively. This survey has also shown a remarkable variation in the rate of the sea level rise and significant acceleration since 1880.

Gregory et al. [13] explored the important issue of the observed sea-level increase exceeding the sum of identifiable contributions in size (especially in the twentieth century). It has been found that the largest contributions come from the thermal expansion of the ocean and the melting of glaciers and ice caps. They proposed several reasons for this, such as the failure of climate models to consider the thermal expansion of the oceans caused by volcanic forcing, the rise of glaciers melting compared to previous estimates, and the almost equal and opposite contribution of groundwater depletion and impoundment.

Marzeion et al. [14] evaluated the anthropogenic and natural effect on the global glacier mass loss, suggesting that anthropogenic forcing, in 1851–2010, was responsible for only $25 \pm 35\%$ of the global glacier mass loss, while in 1991–2010 it increased to $69 \pm 24\%$.

Jevrejeva et al. [15] studied the natural and anthropogenic effects on the sea-level histories (in the last millennium) extracted from a statistical model. As it was derived, the natural signal was the main cause of sea-level variability observed until 1800. In contrast, the observed sea-level rise of the twentieth century was outside the boundaries of its natural variability, and the observed increase in greenhouse gas concentration appears to account for 75% of reported global sea-level trends.

Similar results were derived from other studies. Slangen et al. [16] investigated the variability of sea level based on a climate model, suggesting that natural effects combined with the response to past climatic change are responsible for $67 \pm 23\%$ of the observed sea-level increase before 1950 and only $9 \pm 18\%$ after 1970. In contrast, the anthropogenic signal was responsible for only $15 \pm 55\%$ of the rise in sea level before 1950 but reached $72 \pm 39\%$ in 2000.

Along these lines are the results obtained from the analysis carried out by Marcos et al. [17], where the sea level change (observed in the 20th century) was attributed almost exclusively to anthropogenic causes.

In addition, a few recent studies have attempted to explore scaling dynamics at sea level (i.e., the possibility of long-range dependence, self-similarity, and fractal behavior in the sea-level time series). The term "long-range dependence" describes the property of a quantity so that there are values that remain residually correlated to each other even after many years. The terms "self-similarity" and "fractal" have the meaning that a time-series is exactly or almost like a part of itself. From this point of view, Fraedrich and Blender [18] studied temporal correlations in the surface air temperature over the oceans (i.e., a basic thermodynamic parameter considered to be the main cause of dynamic sea-level behavior). A scaling exponent almost equal to unity was detected, revealing long-term dependence.

Persistent long-range correlations (LRC) were also proposed by Monetti et al. [19], who investigated the scaling properties of the sea surface temperature time series in the Atlantic and Pacific Oceans. The suggested persistence displayed two different types: the short-time type (with a time scale of $\tau < 10$ months), which was characterized by non-stationary behavior for both oceans and the long-term correlation decay. About this, Dangendorf et al. [20] performed a mono-fractal analysis suggesting that sea levels exhibit LRC over time scales up to several decades. They also found long-term correlations to the mass loss from glaciers and ice caps.

Becker et al. [21] studied sea level change over the last century and found that it is beyond its natural internal variability, exhibiting power law long-term correlations on a global and regional scale. Additionally, a power-law scaling behavior was also detected in the long-term sea level variability by Tomasicchio et al. [22]. In this context, Gao et al. [23] recently showed that sea level anomalies exhibit multifractal rather than monofractal behavior.

Multifractal systems are considered a generalization of fractal systems in the sense that their dynamics cannot be described by a single exponent but by a continuous spectrum of exponents.

In conclusion, whereas there's scientific agreement that global and regional mean sea levels have increased since the late 19th century, the relative contribution of natural and man-made forcing remains unclear.

With this in mind, we are trying in the present study to investigate the scaling dynamics in the temporal evolution of the GMSL using two well-known techniques, detrended fluctuation analysis (DFA) and multifractal DFA (MF-DFA), to contribute to the improvement of the future predictions of the sea-level forecasting models, taking into account any systematic rise [24].

## 2. Materials and Methods

For the present analysis, we use two sets of GMSL data (in mm). The first contains monthly mean GMSL measurements over the period 1993–2020, obtained from the TOPEX/Poseidon, Jason-1, Jason-2, and Jason-3 satellite altimeter missions http://www.cmar.csiro.au/sealevel/sl_hist_last_decades.html (accessed on 14 September 2023). The second, which includes monthly mean GMSL values from 1880 to 2013, was downloaded from the website http://www.cmar.csiro.au/sealevel/sl_data_cmar.html (accessed on 14 September 2023), which uses a list of stations with coastal and island sea level measurements. Church et al. [25] presented in detail the careful selection and editing criteria used. In particular, the study of Church and White [12] should be considered along with the Calafat et al. [26] paper, which investigates how well methods based on empirical orthogonal functions can reconstruct global mean sea level [27].

For the study of the LRC and multifractal features of both time series (i.e., the satellite altimeter dataset-SAD and the reconstructed dataset-RD), we employ the DFA and MF-DFA techniques, respectively [28–36].

To analyze the multifractal properties of the time series data, the method we used is MF-DFA, whose steps are presented below:

(1) The first step is to integrate the time series $y(i)$ over time by calculating the differences of the $N$ observations $y(i)$ from their average.

(2) The next step is to divide the integrated time series, $x(i)$, into completely separate boxes of equal length, $\tau$, and repeat the same algorithm starting this time from the end of the profile, thus obtaining $2N_\tau$ boxes (where $N_\tau$ is the integer part of the number $N/\tau$).

(3) The third step is to calculate the polynomial least-square fit (of order $l$) in each box and the corresponding variance obtained from the below formulas (see a more detailed description in [30]):

a. for each box $j = 1, \ldots, N_\tau$:

$$F^2(j, \tau) = \frac{1}{\tau} \sum_{i=1}^{\tau} [x((j-1)\tau + i) - t(i)]^2 \qquad (1)$$

b. for each box $j = N_\tau + 1, \ldots, 2N_\tau$:

$$F^2(j, \tau) = \frac{1}{\tau} \sum_{i=1}^{\tau} [x((N - j - N_\tau)\tau + i) - t(i)]^2 \qquad (2)$$

where $t(i)$ is a locally best polynomial fitted trend (of second degree) to the $\tau$ data.

(4)    In the following, the $q$-th order fluctuation function is estimated by averaging the variances over all boxes:

$$F_q(\tau) = \left[ \frac{1}{2N_\tau} \sum_{j=1}^{2N_\tau} \left[ F^2(j, \tau) \right]^{q/2} \right]^{1/q} \tag{3}$$

where $q$ is the variable moment.

In case of $q \rightarrow 0$, the Equation (3) becomes as follows:

$$F_0(\tau) = \exp\left[ \frac{1}{2N_\tau} \sum_{j=1}^{2N_\tau} \ln\left[ F^2(j, \tau) \right] \right] \tag{4}$$

(5)    The last step is to depict $F_q(\tau)$ vs. $\tau$ (in log-log plot) for different values of $q$ and in the case of multi-scaling behavior, a power-law behavior for $F_q(\tau)$ is observed:

$$F_q(\tau) \sim \tau^{h(q)} \tag{5}$$

where $h(q)$ stands for the generalized Hurst exponent (the slope of the regression line).

Another way to characterize a multifractal series is the singularity spectrum $f(n)$ calculated from $h(q)$ using the modified Legendre transform, where $n$ is the singularity strength or Hölder exponent (see [37]).

It should be remembered that the MF-DFA technique originates from the DFA tool, which evaluates the features of the mono-fractal scaling of a time series [28].

In the case of the original DFA tool, the fluctuation function $F_d(\tau)$ is calculated using Equation (3) for $q = 2$ and first-degree fitted trend (without repeating the algorithm beginning at the end of the profile), i.e.,

$$F_d(\tau) = \left[ \frac{1}{N_\tau} \sum_{j=1}^{N_\tau} \left[ F^2(j, \tau) \right] \right]^{1/2} \tag{6}$$

A power-law behavior of $F_d(\tau)$ (i.e., $F_d(\tau) \sim \tau^a$) is expected for a fractal series, where $a$ is the monofractal exponent. In the case that the $a$-exponent belongs to the interval $(0, 0.5)$, power-law anticorrelations (antipersistence) are detected, while an $a$-exponent value between 0.5 and 1.5 denotes long-range power-law correlations (persistence). For $a = 0.5$, the series is white noise, while for $a = 1$, the series is the $1/f$ noise ($1/f$ noise or pink noise is a process with a frequency spectrum such that the power spectral density is inversely proportional to the frequency of the signal).

It is known that many noisy signals in real systems display trends that deform the scaling results obtained from the DFA method. In this regard, Hu et al. [38] have systematically studied the effects of trends on the DFA results. To avoid such trend interference with our analysis, both GMSL time series (derived from the satellite altimeter dataset and the reconstructed dataset) were initially detrended by applying a polynomial best fit (of sixth-order) and then they were deseasonalized by applying the Wiener filter that produces an estimate of a random process by linear time-invariant filtering [39]. It is worth noting that the sixth-order polynomial fitting gave the most significant results (at a 95% confidence level) without, however, removing the relevant long-term oscillations from the studied time series.

## 3. Results

### 3.1. GMSL Derived from Satellite Altimeter Data

Figure 1 illustrates the temporal evolution of the monthly mean GMSL values obtained from SAD during the period 1993–2020. According to Figure 1, a clear upward trend is revealed, with a magnitude of $3.54 \pm 0.03$ mm per year.

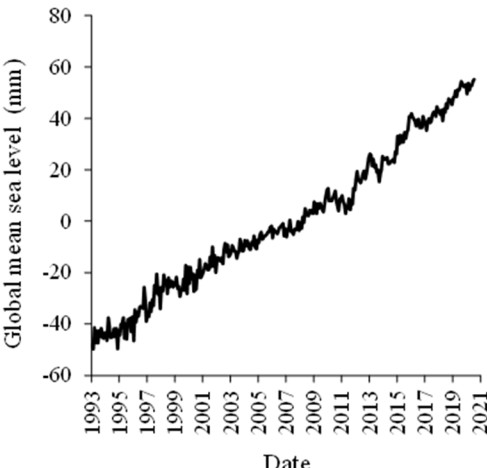

**Figure 1.** Monthly mean values of GMSL from 1993 to 2020 as estimated from SAD.

To investigate in-depth, the temporal course of GMSL as derived from SAD, it is important to examine its possible intrinsic scaling features by detecting whether the sea level at different times is correlated. For this reason, we applied the DFA method to the GMSL (from SAD) time series, and the results obtained are presented in detail in the next section.

#### 3.1.1. Application of the DFA Method to the GMSL Derived from SAD

Figure 2a shows the detrended and deseasonalized (D&D) time series of the monthly mean GMSL values as deduced from SAD. The scaling exponent derived from DFA was $a = 0.77 \pm 0.02$, thus indicating the persistent LRC (see Figure 2b).

It should be noted that 0.02 is the standard error of $a$ which is calculated by the equation: $s_a = \sqrt{\dfrac{\sum\limits_{i=1}^{n} [\log F_d(\tau_i) - y_i]^2}{(n-2)\sum\limits_{i=1}^{n} [\log \tau_i - \overline{\log \tau}]^2}}$ and $\overline{\log \tau}$ is the mean value of $\log \tau_i$, $i = 1, \ldots, n$. It turns out that the *t*-statistic $t = \frac{a}{s_a}$ obeys the Student distribution with $(n-2)$ degrees of freedom. Thus, the 95% confidence interval of $a$ is $(a - s_a t_{n-2,0.025} \,,\, a + s_a t_{n-2,0.025})$ (see Section 7.12.1 of [40]).

To reject the hypothesis that the above-mentioned scaling dynamics could come from random noise, we used DFA in a 900-time series of random (white) noise (i.e., Monte Carlo simulations) of the same size as the SAD one. By this, we will set the 95% confidence interval of $\bar{a}$ (i.e., the average value of $a$). The confidence interval for the white noise had to be constructed to test (with statistical accuracy) the hypothesis $H_0$: $\bar{a} = 0.77$ vs. $\bar{a} \neq 0.77$. According to the non-parametric Kolmogorov-Smirnov test, the extracted $a$-exponents were found to obey a Gaussian distribution at a 95% confidence level with an average $\bar{a} = 0.53$ and a standard deviation $\sigma_a = 0.09$ [41,42]. Therefore, the 95% confidence interval of $\bar{a}$ is:

$$\left(\bar{a} - \frac{1.96}{\sqrt{900}}\sigma_a \,,\, \bar{a} + \frac{1.96}{\sqrt{900}}\sigma_a\right) = (0.52 \,,\, 0.54) \tag{7}$$

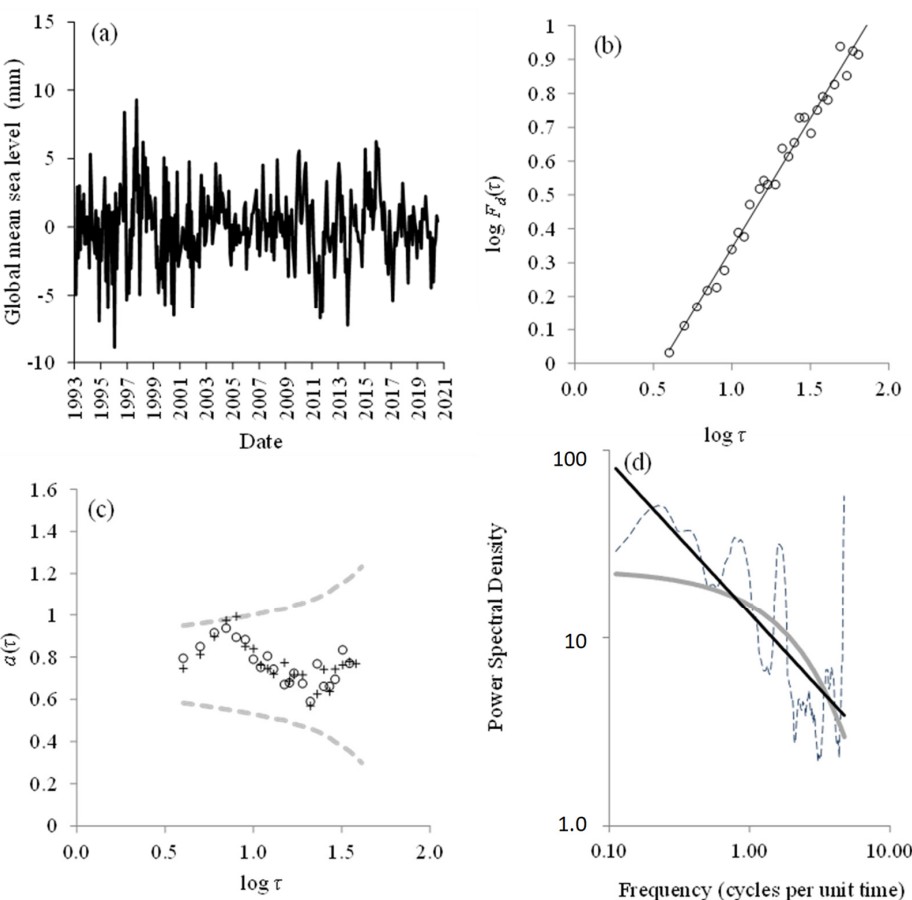

**Figure 2.** (**a**) GMSL time series (derived from SAD) that have been detrended and deseasonalized over the period 1993–2020. (**b**) Root-mean-square fluctuation function $F_d(\tau)$ (derived from the DFA technique) vs. time scale $\tau$ (in months), in a log-log plot, for the time series described in (**a**) (with the best-fit equation: $y = 0.77x - 0.42$ and $R^2 = 0.99$). (**c**) Local slopes $a(\tau)$ vs. $\log\tau$ estimated within a window of 10 and 8 points (circles and crosses). The $2\sigma_{a(\tau)}$ intervals around the mean value $\bar{a} = 0.77$ of the local slopes are shown with the grey line. (**d**) The fit of the power spectral density vs. frequency $f$ (dotted line) with the power-law line (black) and exponential line (grey).

Since the DFA exponent of the GMSL time series (from SAD) (i.e., 0.77) was found to be higher than the upper limit of the interval shown in (7), persistent characteristics were disclosed (see Figure 2b). It is worth mentioning that the interval given in Equation (7) does not contain 0.5, indicating that the DFA exponent for the random noise time series is approximately (but not exactly) equal to the value 0.5. Persistent LRC means that sea-level fluctuations, from short to longer intervals (up to 6.3 years or $\log \tau = 1.88$), are positively correlated throughout the entire time series. It is noteworthy that the upper limit of 6.3 years (which resulted from the DFA formula and is attributed to the total length of the time series) is very close to the upper limit of irregular intervals (3–7 years) in which the natural tropical phenomenon ENSO occurs. It is an oceanic response to purely stochastic atmospheric forcing having climatological impacts in regions far away from the tropical Pacific (i.e., teleconnections) and may be linked to extreme weather conditions (e.g., floods and droughts), changes in the incidence of epidemic diseases (e.g., malaria), severe coral bleaching, civil conflicts, etc. [31,43,44].

However, to establish the aforementioned LRC in the GMSL time series, we should examine the hypothesis that the power-law decay (i.e., $y = 13.53x^{-0.80}$ with a coefficient of determination $R^2 = 0.52$) could describe the power spectral density profile more accurately than the exponential decay (i.e., $y = 23.1e^{-0.43x}$ with $R^2 = 0.40$) (Maraun et al. [45]). The power-law decay means that a relative change in frequency results in a relative change in

spectral density proportional to the negative power of the change, regardless of the initial magnitude of these quantities. Indeed, the F-test showed that the hypothesis $H_0$: $R^2{}_{\text{power law}}$ > $R^2{}_{\text{exponential}}$ vs. $H_1$: $R^2{}_{\text{power law}} \leq R^2{}_{\text{exponential}}$ could not be rejected at the 95% confidence level. Additionally, another criterion should be used to establish the scaling dynamics in the GMSL time series. This is the stability of "local slopes" in a specified range. For this reason, we applied the DFA method to 1000 series of fractional Gaussian noise with $a = 0.8$ (i.e., Monte Carlo simulations) to compute local slopes-$a(\tau)$ vs. $\log\tau$ for each of them in two separate window sizes (of 8 and 10 points) that were sequentially shifted to all computed scales $\tau$. Thus, we have determined an interval $R = \left( \bar{a} - 1.96\sigma_{a(\tau)} \, , \, \bar{a} + 1.96\sigma_{a(\tau)} \right)$ where $\bar{a}$ is the mean value and $\sigma_{a(\tau)}$ is the standard deviation of the 1000 estimated local slopes-$a(\tau)$. Figure 2c shows that in the case of the GMSL time series, the local slopes of $\log F_d(\tau)$ vs. $\log\tau$ revealed stability, and indeed, the local slopes after $\log\tau = 1$ appeared to belong to the $R$ range derived from Monte Carlo simulations.

To have more information about the plausible existence of LRC in the GMSL time series, the next step in our analysis was to investigate the features of the spectrum of singularities for the D&D GMSL time series using the MF-DFA technique.

### 3.1.2. Application of the MF-DFA Method on the GMSL Derived from SAD

The MF-DFA technique provides an estimate of the fluctuation function $F_q(\tau)$ of the $q$-th order for various moments $q$. The application of this technique to the D&D GMSL time series (derived from SAD) revealed the expected power-law scaling behavior (i.e., $F_d(\tau) \sim \tau^a$), which corresponds to straight lines in the log-log plot, on large scales $\tau > 8$ months ($\tau > 12$ months) for all the selected positive (negative) moments $q$ (see Figure 3a) [30,44].

Then, the generalized Hurst exponent $h(q)$ was depicted as a function of $q$-values to confirm the multifractality of the time series examined (see Figure 3b). The fact that the exponent $h(q)$ varies with $q$ and the $h(q)$ values were higher than 0.5 reveals multifractal behavior and persistent LRC for the GMSL time series. It was also noticed that positive $q$-values (i.e., large fluctuations) corresponded to lower $h(q)$-values, which is in line with the usual features of multifractal time series.

The feature of multifractality detected in GMSL might substantially contribute to current scientific knowledge. To provide more information on this feature, we also plotted the singularity spectrum $f(n)$ versus the singularity strength $n$ [30]. The maximum $f(n)$ value corresponds to $q = 0$ and the $f(n)$ values on both sides of the maximum value correspond to positive or negative moments (see Figure 3c).

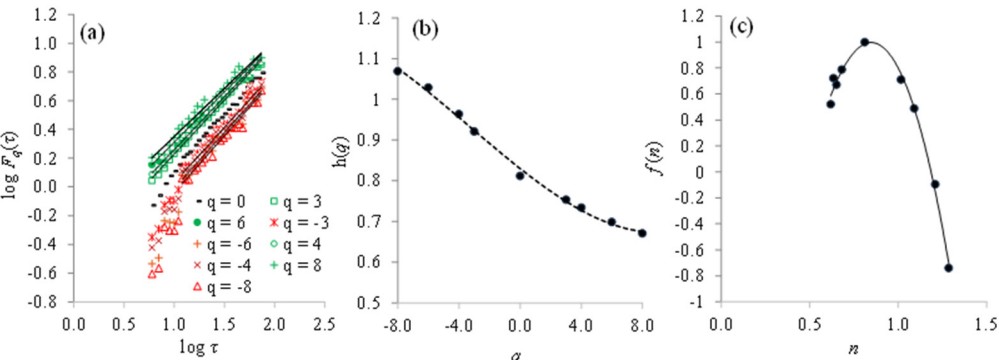

**Figure 3.** (**a**) The MF-DFA fluctuation factor $F_q(\tau)$ as a function of the time scale $\tau$ for various moments $q$ for the D&D GMSL time series (SAD) during 1993–2020. The straight lines in the log-log plot, at large scales $\tau > 8$ months ($\tau > 12$ months) for all selected positive (negative) moments $q$, revealed the expected power-law scaling behavior (i.e., $F_d(\tau) \sim \tau^a$). (**b**) The dependence of the generalized Hurst exponent $h(q)$ on $q$-values for the data shown in (**a**) (for scales 6 months $\leq \tau \leq 6.3$ years). The equation of the best fit is $h(q) = 7 \times 10^{-5}q^3 + 0.001q^2 - 0.03q + 0.83$, with $R^2 = 1.00$. (**c**) The dependence of the singularity spectrum $f(n)$ on the singularity strength $n$. The equation of the best fit is $f(n) = -1.54n^3 - 4.27n^2 + 10.55n - 3.94$, with $R^2 = 0.99$.

It should be clarified and highlighted that the combination of the MF-DFA tool along with the two Maraun criteria [45] is the suitable mathematical methodology to unravel the fractal nature of the evolution of geophysical parameters such as GMSL.

### 3.2. GMSL Derived from Reconstructed Data

To verify the above results, we repeated the same analysis for the mean monthly GMSL values obtained from RD during a longer period (1880–2013). By studying the temporal evolution of this dataset, an apparent upward trend was detected, with a magnitude of $1.6 \pm 0.008$ mm per year (see Figure 4a). Figure 4b depicts the mean monthly GMSL values derived from the SAD over the period 1993–2013 compared to the corresponding part of Figure 4a for the same period.

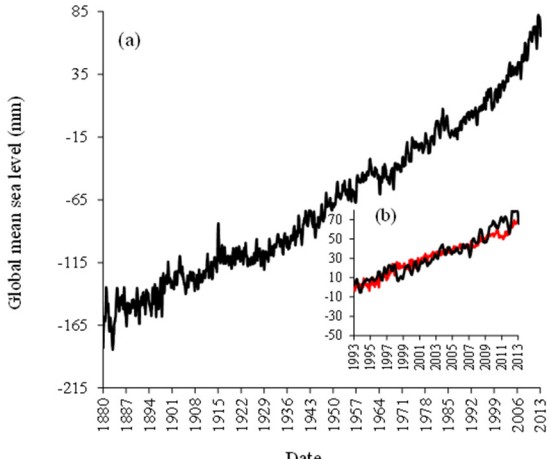

**Figure 4.** (**a**) GMSL mean monthly values from 1880 to 2013 as derived from RD, (**b**) the 1993–2013 part of (**a**) (red line), and the GMSL mean monthly values derived from SAD for the same period (black line).

We then attempted to investigate the intrinsic properties of the GMSL (from RD) time series, applying the DFA technique.

### 3.2.1. Application of the DFA Method on the GMSL Data Derived from RD

The time series of monthly GMSL mean values (from RD) was initially D&D by applying the techniques described in Section 2 (see Figure 5a).

The results of applying the DFA technique to the D&D GMSL time series are shown on a log-log graph in Figure 5b with the corresponding best-fit equation: $y = 0.76x + 0.002$ and $R^2 = 0.97$. This gives a scaling exponent of $a = 0.76 \pm 0.02$. Moreover, the existence of a power-law scaling and LRC in the GMSL (from RD) time series was established by detecting the type of power spectral density and the stability of the local slopes on long-term scales [45].

Figure 5c depicts the local slopes of $\log F_d(\tau)$ vs. $\log \tau$, separately for two separate window sizes of 10 and 12 points, which were shifted successively to all computed scales $\tau$. It is worth noting that these two window sizes were chosen to give the most significant results for the estimated scaling exponents (at the 95% confidence level). As can be seen, the entire local slopes (after $\log \tau = 1.00$) are within the boundary of the $R$ range (determined by the 1000 Monte Carlo simulations), indicating sufficient stability. Figure 5d shows the profile of the power spectral density for the D&D GMSL (from RD) time series, suggesting the rejection of exponential decay ($y = 165.1 \mathrm{e}^{-1.18x}$ with $R^2 = 0.57$) compared to the power law ($y = 29.35 \cdot x^{-1.96}$ with $R^2 = 0.65$), as the hypothesis $H_0$: $R^2_{\text{power law}} > R^2_{\text{exponential}}$ vs. $H_1$: $R^2_{\text{power law}} \leq R^2_{\text{exponential}}$ could not be rejected (using $F$-test at 95% confidence level).

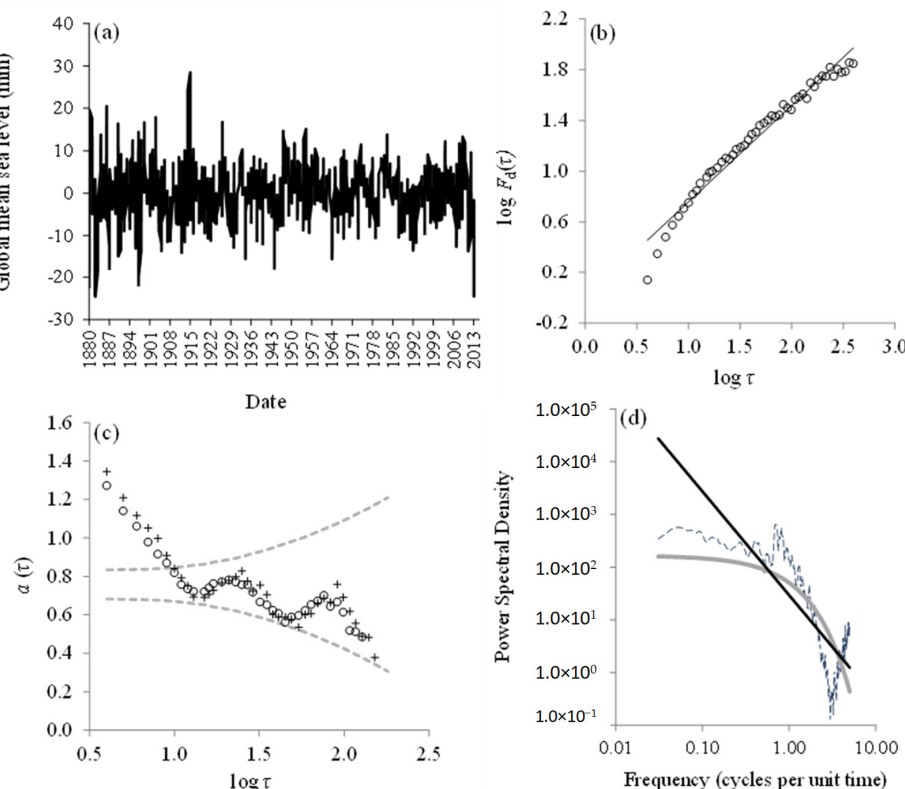

**Figure 5.** As in Figure 2, but for the case of the detrended and deseasonalized GMSL time series (derived from RD) during 1880–2013 (**a**). Panels (**c,d**) are the same as in Figure 2, i.e., the local slopes $a(\tau)$ vs. $\log\tau$ estimated within a window of 10 and 8 points (circles and crosses) with the grey line depicting the $2\sigma_{a(\tau)}$ intervals around the mean value = 0.76 and the fit of the power spectral density vs. frequency $f$ (dotted line) with the power-law line (black) and exponential line (grey), respectively. The best-fit equation in (**b**) is: $y = 0.76x + 0.002$ and $R^2 = 0.97$. The local slopes of $\log F_d(\tau)$ vs. $\log\tau$ shown in (**c**) are calculated in the windows of 12 and 10 points (circles and crosses).

Therefore, the GMSL (from RD) time series meets both criteria proposed by Maraun et al. [45]. However, to study the multifractality and the power-law long-term persistence of the time series considered, we also applied the MF-DFA technique. The results obtained are given in the following section.

3.2.2. Application of the MF-DFA Method on the GMSL Data Derived from RD

The MF-DFA technique was applied to the D&D GMSL time series (derived from RD) and indicated the expected power-law scaling behavior (i.e., $F_d(\tau) \sim \tau^a$) which corresponds to straight lines in the log-log plot, on large scales $\tau > 17$ months for all the selected positive and negative moments $q$ (see Figure 6a).

Additionally, the plot of the generalized Hurst exponent $h(q)$ as a function of $q$ (Figure 6b) showed $h(q)$ values clearly higher than 0.5 and dependence of $h(q)$ on $q$, thus verifying multifractality and the persistent LRC of the time series examined. Also, it is noteworthy that the lower values of $h(q)$ for the positive moments (compared to the negative ones) were consistent with the usual features of the multifractal time series (see Figure 6b).

Finally, we plotted the singularity spectrum $f(n)$ versus the singularity strength $n$ for the specific time series. According to Figure 6c, the maximum $f(n)$ value is around $q = 0$ and the $f(n)$ values on both sides of the maximum value correspond to positive or negative moments (see Figure 6c).

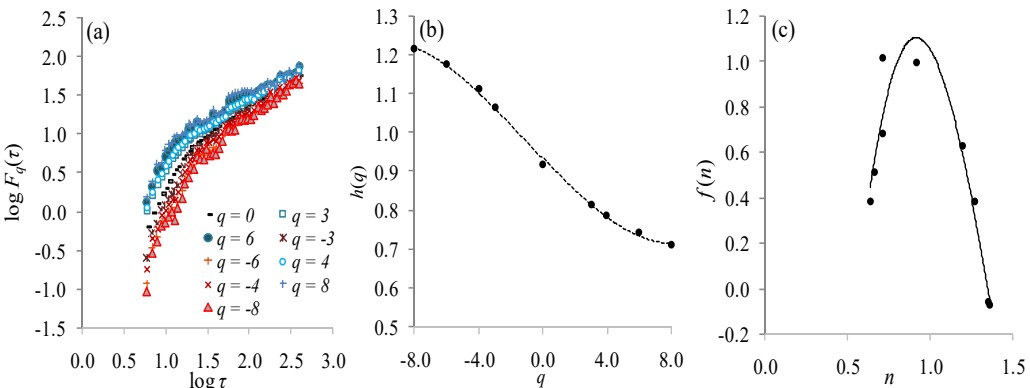

**Figure 6.** As in Figure 3, but for the case of the detrended and deseasonalized GMSL time series (derived from RD) during 1880–2013 (for scales 17 months $< \tau \leq$ 33 years) (**a**). The equation of the best-fitted line in (**b**) is $h(q) = -2 \times 10^{-4} \cdot q^3 + 5 \times 10^{-4} q^2 - 0.04q + 0.94$, with $R^2 = 1.00$, while in (**c**) is: $f(n) = 2.58n^3 - 14.43n^2 + 20.06n - 7.18$, with $R^2 = 0.96$.

### 3.2.3. Application of the MultiFractal Centered Moving Average (MFCMA) Method

As an additional check on the findings presented above, we employ an additional modern multifractal methodology termed MultiFractal Centered Moving Average (MFCMA). MFCMA was introduced by Schumann and Kantelhardt [46], and the details of its implementation are elaborated in their Section 2.1.

MFCMA, which is less computationally demanding than MFDFA, is based on the Centered Moving Average (CMA) method [47,48]. CMA is slightly better than DFA in the limits of small ($\tau < 10$) and large scales ($\tau > N/4$) which, according to Schumann and Kantelhardt [46], makes MFCMA suitable for short time series, which is our present case. Of course, MFCMA performs better for time series in the absence of trends, and for this reason, it was applied to the cases of the detrended and deseasonalized time series shown in Figures 2a and 5a for SAD and RD, respectively.

The results obtained (see, e.g., Figure 7 for RD) lead to singularity spectra comparable to those obtained by MFDFA, with maxima of the singularity spectra at approximately $n = 0.83 \pm 0.02$ and $0.86 \pm 0.01$, for SAD and RD, respectively. The generalized Hurst exponents $h(2)$ result in $0.80 \pm 0.02$ and $0.81 \pm 0.04$, for SAD and RD, respectively. We observe that the results found by DFA and MFDFA are validated by CMA and MFCMA.

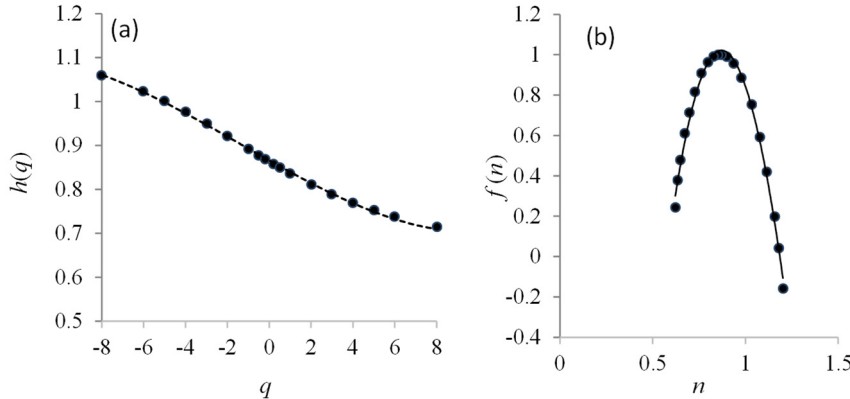

**Figure 7.** As in Figure 3, after applying MFCMA to the detrended and deseasonalized GMSL time series (derived from RD). The equation of the best-fitted line in (**a**) is $h(q) = 8 \times 10^{-5} q^3 + 0.000 q^2 - 0.026q + 0.866$ with $R^2 = 1.00$, while in (**b**) is: $f(n) = 2.14n^3 - 16.63n^2 + 24.12n - 8.780$ with $R^2 = 0.99$.

### 3.3. GMSL Derived from Reconstructed and Satellite Data, during the Common Period

The final step of this study was to examine whether the historical data set could be used in any way to validate the SAD values. For this purpose, we focused on the common

time period of the two datasets (i.e., January 1993–December 2013), and we compared their trends and their scaling properties.

The GMSL time series from SAD and RD showed statistically significant upward trends of $3.18 \pm 0.04$ mm per year and $3.56 \pm 0.06$ mm per year, respectively (see Figure 4b).

Additionally, the application of the DFA technique (during the aforementioned common time period) to the D&D GMSL obtained from SAD and RD gave persistent scaling with exponents $a = 0.73 \pm 0.03$ and $a = 0.78 \pm 0.03$, respectively. These scaling features were established using the criteria of rejection of the exponential decay of the autocorrelation function and the stability of local slopes. Furthermore, the two $a$-values indicate similar scaling properties, suggesting that fluctuations in sea level, from short time intervals to longer ones (up to 5 years), are positively correlated.

As mentioned in the introduction, the long-term persistence of GMSL has already been evaluated using similar approaches [20,21]. However, as noted by Dangendorf et al. [49], none of the current GMSL reconstructions properly represent the temporal variability on time scales up to decades.

To elaborate more on the interpretation of the results presented above, it should be emphasized that MSL variability is closely related to that of sea surface temperature (SST), which displays persistence [19]. MSL also exhibits long-term correlations [20] with changes that are beyond its natural internal variability [20,21]. However, the relationship between SST and MSL is complex because SST is one of many factors that affect MSL, including winds, currents, river discharges, and gravity fluctuations.

Furthermore, the analysis showed that the long-term correlations are mainly due to natural causes, due to the almost identical scaling characteristics for both the 134-year time series and the latest 28-year GMSL time series. This conclusion is only partially consistent with sea-level change being unnatural in two-thirds of the longest tidal records [21].

It is important to emphasize at this point that although time series with strong persistence may exhibit a large upward trend for natural reasons, we cannot rule out anthropogenic forcing (e.g., in the case of a natural downward trend) [21].

Therefore, as also noted by Baker et al. [21], the investigation of the responsible possible combination of external factors for the observed sea level changes requires further research and, indeed, an increase in the available experimental data.

## 4. Conclusions

The main findings of the analysis presented above are as follows:

1. Applying the DFA technique to the D&D GMSL time series from the satellite altimeter dataset (reconstructed dataset) during the period 1993–2020 (1880–2013) gives a scaling exponent $a = 0.77 \pm 0.02$ ($a = 0.76 \pm 0.02$), thus revealing that the fluctuations in mean sea-level values from short to longer time intervals are positively correlated.
2. The application of the MF-DFA technique to both GMSL time series used suggested the power-law scaling behavior of $F_d(\tau)$ on large scales $\tau$ for all the selected positive and negative moments. Additionally, the generalized Hurst exponent $h(q)$ appears to depend on $q$, and the $h(q)$ values were higher than 0.5, revealing multifractality and persistent long-range correlations.
3. A comparison of the trends and scaling properties of both GMSL time series was carried out for the common period (i.e., January 1993–December 2013). Similar scaling properties were revealed for the two-time series, thus suggesting that the historic data set could be used in any way to validate the satellite altimeter dataset.

The above-mentioned multifractality features detected in the GMSL may contribute to the hot topic of projections of global and regional mean sea-level rise and help to integrate a holistic insight for devising ad hoc strategies and mitigating inevitable impacts [50,51]. In relation to this, under accelerating sea-level rise, it is currently impossible to determine the future evolution of climate, marshes and mangroves as, for example, salt marshes are capable of laterally expanding, contracting, and vertically accumulating in response to sea-level rise [52–54].

**Author Contributions:** Conceptualization, C.V. and Y.M.; methodology, N.V.S., M.E. and C.V.; software, N.V.S. and M.E.; validation, C.V., N.V.S. and M.E.; formal analysis, M.E. and N.V.S.; investigation, C.V., Y.M. and M.E.; resources, Y.M. and D.S.; data curation, M.E.; writing—original draft preparation, C.V. and M.E.; writing—review and editing, Y.M., N.V.S. and D.S.; supervision, C.V. and Y.M. All authors have read and agreed to the published version of the manuscript.

**Funding:** This research received no external funding.

**Data Availability Statement:** The data presented in this study are available on request from the corresponding author.

**Conflicts of Interest:** The authors declare no conflict of interest.

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
