# Peer review of "On the Impacts of the Global Sea Level Dynamics"

_fractalfract, doi:10.3390/fractalfract8010039_

Round 1
Reviewer 1 Report
Comments and Suggestions for Authors
The authors use two well-established techniques to study the scaling properties of the global mean sea level time-series. They show that there are long-range correlations in the datasets, and suggest that these stem from natural causes. The methodology was sound and the importance of the topic was well-highlighted, therefore I propose the publication of this paper after some minor revisions below:
a) pg 3, lines 116-124 are leftover text from the template and must be removed.
b) pg 6, line 229. Some explanation is needed for this form of power-law decay
c) pg 8, line 295. What is the significance of these window sizes? How were they chosen? Do the results vary significantly if these are changed?
Author Response
Please see the file attached.

Reviewer 2 Report
Comments and Suggestions for Authors
Dear Authors,
This study, while addressing an important topic, in my opinion, lacks the required originality in the implemented methodologies, making it unsuitable for publication in this Journal, which has the Mathematics, Interdisciplinary Applications category.
I have outlined some specific minor comments:
- The data presented in the manuscript only extends up to the year 2020. Is it feasible to extend the time period?
- Regarding the (MF-)DFA methodology, consider presenting the steps more organized –for example, enumerate steps.
- Ensure the use of standard notation throughout the manuscript.
- To further enrich the analysis of your results, I recommend including more multifractal metrics. This expansion could provide a more comprehensive understanding of the complexities within your data.
Best regards,
Reviewer
Comments on the Quality of English Language
The quality of English seems acceptable.
Author Response
Please see the file attached.

Reviewer 3 Report
Comments and Suggestions for Authors
In their manuscript, Varotsos et al. employ detrended fluctuation analysis (DFA) for investigation of scaling properties of two versions of global sea level series (covering periods 1993-2020 and 1880-2013, respectively). The results are found to be similar for both versions of the series, confirming presence of long-range correlations in these signals and their robustness over different periods of climate history.
Aim of the analysis fits well into the Fractal and Fractional journal. There are, however, some aspects of the methodology documentation and presentation of the results that could still be improved a bit (please, see below for details). Furthermore, I feel it would be useful to elaborate more on interpretation of the results and their relation to previous studies dealing with a similar problem, particularly papers by Monetti et al. (2003), Dangendorf et al. (2014) and Becker et al. (2014) (referenced as [19, 20, 21] in the current manuscript). More specifically:
Main comments:
(C1) l. 152: What specific form of polynomial was used in (MF)FDA?
(C2) l. 183: The time series were pre-processed by detrending based on 6-th order polynomial. Why was this specific order used? (it seems quite high to me - I wonder if some of the relevant long-term oscillations may be removed from the sea level series this way)
(C3) l. 200: What method was used to create the confidence interval?
(C4) 2nd paragraph of Sect. 3.1.1: Is it necessary to construct the confidence interval for white noise? The value should be 0.5 automatically, as mentioned at l. 175; you can then compare it to the confidence interval in the previous paragraph. (BTW, it is also curious that the interval given in eq. (7) does not actually contain 0.5)
(C5) l. 19-21 (Abstract): ‘In addition, the analysis revealed nearly identical scaling features for both the 134-year and the last 28-year GMSL-time series, possibly suggesting that the long-range correlations stem more from natural causes.’: It would be useful to discuss this more deeply in the text, especially in the context of outcomes of previous similarly focused studies. For instance, one of the highlights of the paper by Becker et al. (2014) ([21] in the current manuscript) reads ‘Sea level change is clearly unnatural in two thirds of the longest tidal records’ – how well does this conclusion align with the findings of the current manuscript, in terms of influence of major components to sea level variability (such as trends vs. long-term oscillations vs. short-term oscillations)?
Minor/technical comments:
(M1) Some figures use decimal comma rather than decimal dot.
(M2) l. 239: Just a detail, but sometimes a factor of 2 is used to construct the 95% confidence intervals, sometimes 1.96 is used (e.g. in eq. (7)).
(M3) l. 251: ‘... which corresponds to straight lines in the log-log plot, ...’: Maybe these lines could be added to Fig. 3a.
Comments on the Quality of English Language
I detected no major problems with English of the manuscript.
Author Response
Please see the file attached.

Round 2
Reviewer 2 Report
Comments and Suggestions for Authors
Dear Authors,
I appreciate the enhancements you've incorporated into the paper; they have positively impacted my assessment. While my initial impression was that the methodology might not be deemed innovative for this journal, the nature of the paper's topic leads me to recommend its acceptance. I have no further remarks.
Best regards,
Reviewer
Author Response
Dear Reviewer
The improvements we incorporated into our paper following your suggestions have strengthened it considerably. We therefore believe that your input was valuable.
With my best season's wishes
Costas VAROTSOS
Reviewer 3 Report
Comments and Suggestions for Authors
The authors have largely addressed my comments (though the method of confidence interval construction is still not given for the estimate at l. 201), I believe the paper can be published now.
Comments on the Quality of English Language
I found no major problems with English of the paper.
Author Response
Please see the attached file.
Thank you and best season's wishes
